# Complementary Feeding and Overweight in European Preschoolers: The ToyBox-Study

**DOI:** 10.3390/nu13041199

**Published:** 2021-04-05

**Authors:** Natalya Usheva, Sonya Galcheva, Greet Cardon, Marieke De Craemer, Odysseas Androutsos, Aneta Kotowska, Piotr Socha, Berthold V. Koletzko, Luis A. Moreno, Violeta Iotova, Yannis Manios

**Affiliations:** 1Department of Social Medicine and Health Care Organization, Medical University of Varna, 9002 Varna, Bulgaria; 2Department of Pediatrics, Medical University of Varna, 9002 Varna, Bulgaria; sonya_galcheva@mail.bg (S.G.); iotova_v@yahoo.com (V.I.); 3Department of Movement and Sports Sciences, Ghent University, 9000 Ghent, Belgium; greet.cardon@ugent.be; 4Department of Rehabilitation Sciences, Ghent University, 9000 Ghent, Belgium; marieke.decraemer@ugent.be; 5Research Foundation Flanders, 1000 Brussels, Belgium; 6Department of Nutrition and Dietetics, School of Physical Education, Sport Science and Dietetics, University of Thessaly, 382 21 Volos, Greece; oandrou@hua.gr; 7Public Health Department, Children’s Memorial Health Institute, 04-730 Warsaw, Poland; A.kotowska@ipczd.pl (A.K.); p.socha@ipczd.pl (P.S.); 8Division of Metabolic and Nutritional Medicine, Department Paediatrics, Dr. von Hauner Children’s Hospital, LMU University Hospitals, 80337 Munich, Germany; berthold.koletzko@med.uni-muenchen.de; 9GENUD (Growth, Exercise, Drinking Behaviour and Development) Research Group, University of Zaragoza, 50009 Zaragoza, Spain; lmoreno@unizar.es; 10Department of Nutrition and Dietetics, Harokopio University, 176 76 Athens, Greece; manios@hua.gr

**Keywords:** complementary feeding, solid food, breastfeeding, overweight, obesity

## Abstract

Complementary feeding (CF) should start between 4–6 months of age to ensure infants’ growth but is also linked to childhood obesity. This study aimed to investigate the association of the timing of CF, breastfeeding and overweight in preschool children. Infant-feeding practices were self-reported in 2012 via a validated questionnaire by >7500 parents from six European countries participating in the ToyBox-study. The proportion of children who received breast milk and CF at 4–6 months was 51.2%. There was a positive association between timing of solid food (SF) introduction and duration of breastfeeding, as well as socioeconomic status and a negative association with smoking throughout pregnancy (*p* < 0.005). No significant risk to become overweight was observed among preschoolers who were introduced to SF at 1–3 months of age compared to those introduced at 4–6 months regardless of the type of milk feeding. Similarly, no significant association was observed between the early introduction of SF and risk for overweight in preschoolers who were breastfed for ≥4 months or were formula-fed. The study did not identify any significant association between the timing of introducing SF and obesity in childhood. It is likely that other factors than timing of SF introduction may have impact on childhood obesity.

## 1. Introduction

Obesity is an increasing worldwide problem with an estimate of 340 million overweight or obese children and adolescents aged 5–19 in 2016 and 38.2 million children under the age of 5 years being overweight or obese worldwide in 2019. Moreover, health expenditures for the adult population are constantly increasing, €70 billion per year in Europe (2017) and $342.2 billion in the US (2013). Direct medical costs related to childhood obesity alone were approximately $14 billion in 2013 and they are expected to rise significantly, especially because today’s obese children are likely to become tomorrow’s obese adults [1,2,3,4,5].

One of the risk factors for childhood obesity is inappropriate nutrition during infancy. The advantages of exclusive breastfeeding (EBF) compared to partial breastfeeding in the first months of life have been recognized. The World Health Organization’s global public health recommendations promote “exclusive breastfeeding for 6 months” with continued breastfeeding up to the age of two years or beyond. Complementary feeding (CF) should occur when a baby is both developmentally ready and when breast milk is no longer able to fulfil the nutritional requirements of the child [6,7]. The recommended period for starting CF as stated by the European Society for Pediatric Gastroenterology, Hepatology and Nutrition is at week 17–26 (between the beginning of the fifth month and the beginning of the seventh month of life) [8].

The protective role of breastfeeding (BF) against overweight and obesity has been reported in many studies, showing a greater effect with longer duration of breastfeeding [9,10,11]. However, the association of timing of CF introduction and the quantity and quality of CF with childhood obesity is controversial [12,13,14,15,16,17,18]. Some studies reported an association of very early CF introduction before four months of age with later obesity in formula-fed infants whereas there was little effect in breastfed infants [19]; overweight and obesity at 2–12 years; obesity at three years [20,21]. An association between early CF introduction and overweight in children aged 1–17 has also been reported as being modified by the duration of breastfeeding in a birth cohort study in the Netherlands (Prevention and Incidence of Asthma and Mite Allergy—PIAMA) [22]. These findings are not consistent with the conclusions of the systematic review by Pearce et al. who found no consistent association between very early introduction of CF (prior to the age of four months) and childhood body mass index (BMI) [23].

Since there is no consensus yet on a possible relationship between introduction of solid foods (SF) and overweight in childhood, the aim of this study was to investigate the association between the timing of CF, breastfeeding status and overweight in a large pan-European sample of preschool children.

## 2. Materials and Methods

The ToyBox-study was conducted between May and June 2012 in six European countries (Belgium, Bulgaria, Germany, Greece, Poland and Spain) among parents/caregivers of preschoolers born between January 2007 and December 2008. The ToyBox-study (www.toybox-study.eu; accessed on 20 September 2020) adhered to the Declaration of Helsinki and the conventions of the Council of Europe on human rights and biomedicine. All the countries (Belgium, Bulgaria, Germany, Greece, Poland and Spain) obtained ethical clearance from the relevant ethics committees and local authorities and all parents/caregivers provided a signed consent form before being enrolled in the study. Detailed information about the ToyBox-study design has been previously reported [24]. 

Data about perinatal information of preschoolers (including anthropometric measurements at birth, breastfeeding and complementary feeding practices during the first year of life) were obtained through a standardized self-administered questionnaire for primary caregivers. They also aimed to include sociodemographic characteristics of the participants. The questions about children’s nutrition during the first year of life were formulated with a focus on the presence/absence of breastfeeding at each month after birth and the age at which water, tea, juice, formula milk and solid/semi-solid foods were introduced. For the purpose of limiting the recall bias, parents/caregivers were advised to use the child’s medical records for the questions in the perinatal section, resulting in an excellent value of ICC (intraclass correlation coefficient) in the test–retest reliability study—0.75, whereas the questions on parental weight and height had “moderate-to-excellent reliability” (i.e., ICC ranged from 0.489 to 0.911) [25]. Data on health-related behaviors (dietary habits, physical activity and sedentary behavior) of preschoolers and their parents were collected using validated questionnaires and the results regarding this topic are presented in other papers [26,27].

Family socioeconomic status (SES) was categorized according to maternal years of education as “low SES” (≤12 years), “medium SES” (13–16 years) and “high SES” (≥16 years of education). Preterm birth was defined as <37 gestational weeks, full-term birth—as ≥37 gestational weeks. Breastfeeding status was defined according to the WHO indicators [28]. Exclusive breastfeeding indicated breastfeeding with no other food or liquid given, except for medical drops and syrups (vitamins, minerals, medicines). Predominant breastfeeding applied if the infant received additional water or water-based liquids. The inclusion of other milks and foods (formula milk and/or semi-solids) was considered partial breastfeeding. The ever breastfed rate was the proportion of infants aged less than 12 months who were ever breastfed. Complementary feeding included liquids and SF (fruit juice, fruits, vegetables, meat, fish, eggs, milk products, creams and soup). Timely complementary feeding rate was defined as the proportion of infants 4–6 months of age who received breast milk and CF. Children without any information about feeding in the first two months (*n* = 454) and without information about the time of solid foods’ introduction (*n* = 300) were excluded from the analyzed study sample (*n* = 7554). Both the analyzed and the excluded samples have similar distribution by country and participating status (intervention or control groups).

### 2.1. Anthropometric Data

Children’s weight (to the nearest 0.1 kg) and height (to the nearest 0.1 cm) were measured using a standardized protocol and standardized equipment which was calibrated before and during the period of data collection [29]. All measurements were taken by research assistants who were thoroughly trained before the initiation of the study to achieve very good intra- and interobserver reliability agreement [30]. Overweight including obesity was defined on the basis of the WHO criteria as BMI z-score > 2 standard deviations (SD) and BMI z-score > 3 SDs, respectively, for children aged < 5 years. For children aged > 5 years, overweight was defined as BMI z-score > 1 SD and obesity as BMI z-score > 2 SDs. Calculation of the ponderal index (PI = weight/height^3^) was used for assessment of the weight status of children at birth, with a PI range of 2.0–3.0 g/cm^3^ considered normal. Children with a PI > 3.0 were considered overweight, and those with PI < 2.0 were classified as small for gestational age (SGA). Parental weight and height were self-reported by parents/caregivers and their BMI was calculated. Parents/caregivers were categorized according to their BMI as “under-/normal weight” (≤24.9 kg/m^2^), “overweight” (≥25 and ≤29.9 kg/m^2^) or “obese” (≥30 kg/m^2^) [31].

### 2.2. Statistical Analyses

Normal distribution of variables was tested with Shapiro–Wilk tests. Continuous variables are presented as the means ± standard deviation in case they were normally distributed (e.g., age of preschoolers, age of mothers, introduction of solid foods) and as the medians and IQR (interquartile range) for non-normally distributed variables (duration of breastfeeding, introduction of tea, introduction of fruit juices). Statistical analysis of parameters’ distribution of the original samples by country was not applied as their number was small (*n* = 6). Post-sampling or bootstrapping were not considered. 

Categorical variables were analyzed using the χ^2^ test regarding country and children’s BMI categories. An independent samples *t*-test was applied for comparison of the means and percentages from two samples while the one-way ANOVA (analysis of variance) for the means of more than two samples (birth weight; mother’s age).

Comparison of the medians was performed using the median test. The association of feeding practices and children’s BMI as well as mother’s characteristics was determined by the correlation analysis. Logistic regression analysis with 95% confidence intervals (CIs) was used to estimate the odds of being overweight/obese (dependent variable) in relation to different infant-feeding practices. The results were adjusted for mother’s age and BMI before pregnancy, SES, smoking habits during pregnancy and country. In order to quantify the probability of complying with current recommendations for the introduction of CF at 4–6 months of age, logistic regression analysis was performed and adjusted for mother’s age and BMI before pregnancy, SES, smoking habits during pregnancy and country. Compliance to recommendations for introduction of CF at 4–6 months of age (yes/no) was considered as a dependent variable. 

In the logistic regression models, the variables were selected based on their relevance for the research topic and being tested for absence of collinearity, hence the presented model coefficients correspond to variables with no significant impact as well. Thus, we can reach a conclusion about the existence of meaningful links. The regression analyses of the current data were targeted at identification of the relevant links between the study variables, but not at constituting a universal model which may be applied for analysis of other populations or for establishing new theories. The data were analyzed using the Statistical Package for Social Sciences (IBM SPSS v. 20, Chicago, IL, USA). The level of significance was set at *p* < 0.05.

## 3. Results

The total number of analyzed eligible questionnaires from the six countries was 6800 (mean age of the participants, 4.75 ± 0.43 years; 47.7% girls with no statistically significant difference in gender distribution between the participating countries). The sociodemographic characteristics of responders are presented in Table 1. Additional information regarding characteristics of the ToyBox-study sample were previously reported [24,31].

Tea (chamomile and other types of tea, especially for baby colics) was the first CF for most of the children in our study, introduced at a median age of three months (IQR, 2–5 months), resulting in a low proportion of exclusively breastfed children at four months of age (Table 2). 

In the study sample, the median introduction to fruit juices was at six months of age (IQR, 5–8 months), with the earliest introduction being among Bulgarian children (median, four months of age; IQR, 3–6 months). In the total sample, the proportion of 4–6-month-old infants who received breast milk and CF (timely complementary feeding rate) was 51.2% (35.4%, Belgians; 50.4%, Spanish; 53.7%, Polish; 67.8%, Bulgarians; 72.3%, Germans (*p* < 0.01)). The median age of SF introduction was six months (IQR, 5–6 months), with the earliest introduction in Belgium (median, four months of age, IQR, 4–5 months). Some 26.6% (*n* = 1806) introduced CF outside of the recommended age range, with 4.1% (*n* = 279) before the 16th postnatal week and 22.5% (*n* = 1527) after the 25th week. The time of CF introduction was correlated with breastfeeding duration (Spearman’s ρ = 0.2; *p* < 0.001). There was a weak positive relationship between the introduction of CF and SES (Spearman’s ρ = 0.08; *p* < 0.001) and a negative relationship with smoking during pregnancy (Pearson’s r = –0.04; *p* = 0.003). Stratifying by country, a negative relationship with smoking during pregnancy was identified only in the Bulgarian sample (Spearman’s ρ = 0.12; *p* < 0.001). 

The prevalence of overweight and obesity according to the WHO I criteria was 8.0% (*n* = 542) and 2.8% (*n* = 190), respectively (Table 1). Infant-feeding practices showed a different relationship to the prevalence of overweight and obesity at different stages of childhood. Timely introduction of CF at 4–6 months of age had a negative association with the prevalence of overweight and obesity at six and 12 months of age, with no differences between breastfed and non-breastfed children (*p* < 0.05) (Table 3).

On the country level, the significant difference in the prevalence of overweight and obesity at preschool age was observed only in two countries—late CF introduction (after seven months) compared to earlier introduction is related to higher prevalence of obesity in Belgium (*p* < 0.001) and to higher prevalence of overweight in Poland (*p* < 0.05) (Table 4).

Table 5 presents results of the logistic regression analysis identifying one single risk factor connected to inappropriate timing of CF introduction (<4 months of age or >6 months of age)—lower SES (OR = 1.25; 95% CI, 1.08–1.45). 

The logistic regression analysis showed that the odds of becoming overweight at preschool age among children who had early introduction of SF (1–3 months of age) compared to those with CF introduction at 4–6 months of age was 0.69 (OR = 0.69; 95% CI, 0.41–1.16; *p* = 0.16). The children introduced to SF before four months of age had a trend for a different overweight risk at preschool age according to their BF status which was not significant when adjusted for the mother’s characteristics (SES, education, pre-pregnancy weight and smoking habits during pregnancy). For the children breastfed for ≥4 months, early introduction of CF was associated with a trend for higher later overweight (OR = 1.23; 95% CI, 0.29–5.14; *p* = 0.78), while among the exclusively formula-fed breastfed children, this risk tended to be lower (OR = 0.39; 95% CI, 0.052–3.12; *p* = 0.38). Late CF introduction at 7–12 months of age was not related to a difference in later overweight and obesity risk when adjusted for country, age and gender (OR = 0.98; 95% CI, 0.83–1.21; *p* = 0.99). 

## 4. Discussion

Our study investigated the association between the timing of CF, breastfeeding status and overweight among European preschool children. Breastfed children (any type of breastfeeding) throughout the first 4–6 months of life and after the 12th month had a lower prevalence of overweight/obesity in childhood compared to formula-fed children. This finding is consistent with the European studies reporting similar results in previous cohorts [9,14,32,33]. The main findings point at a lower prevalence of overweight/obesity at six months of age (*p* < 0.001) in children with introduction of solid foods between 4–6 months of age compared to late introduction (7–12 months of age). However, there were no significant findings for the prevalence at one year and at preschool age (*p* > 0.05). A late SF introduction is related to a higher prevalence of overweight and obesity at six months of age, 12 months of age and at preschool age. Our results are consistent with the previously reported findings [15,18]. 

Pluymen et al. and Huh et al. reported that the duration of breastfeeding modifies the association between CF introduction and overweight: BF for less than four months and CF introduction before four months of age increased the risk for overweight by 37% compared to those with CF introduction ≥ 4 months of age [21,22]. We found a non-significant trend for an association of early introduction of SF and preschool overweight in breastfed children but not in formula-fed children. 

Different previous studies aimed at identification of predictors of children’s dietary intake such as SES and geographic region [34,35]. SES is one of the most commonly identified factors associated with childhood overweight and obesity and reflects a child’s living conditions. However, there is uncertainty as to the mechanisms through which SES influences the child’s weight. Breastfeeding practices and timing of CF introduction are related to SES and other maternal characteristics such as BMI, age at birth, tobacco use during pregnancy, gestational weight gain, depression and use of day care [36,37,38,39]. Results from the ToyBox-study show that mothers with low SES are more likely to have overweight/obese children compared to those with medium/high SES (OR = 1.41; 95% CI, 1.17–1.71 [31].

The current analysis supports the findings of other studies that significant risk factors associated with non-compliance to the recommendation for introduction of CF at 4–6 months of age are low SES and smoking throughout pregnancy (*p* < 0.05) [14,40,41,42]. Maternal educational level did not modify the association of CF < 4 months of age and overweight in the PIAMA cohort as well [19]. Rose et al., based on the data of the Infant Feeding Practices Study II and Year 6 Follow-Up Studies, suggested that the mother’s decisions about milk-feeding and the types and quality of solid foods introduced in infancy can shape dietary patterns and obesity risk later in childhood. Infants who were offered foods high in energy density at nine months of age had a higher intake of these foods at six years of age and a higher prevalence of overweight compared to other classes of dietary patterns [43]. 

Our results, which hopefully will be useful for improving effectiveness of childhood obesity prevention programs in Europe, can also be utilized in low developed and developing countries. Although malnutrition is still a major challenge across the African continent, the largest growth of obesity among 5- to 19-year-olds in the world between 1975 and 2016 was observed in southern Africa (about 400% per decade) [44].

A methodological limitation of the report is the cross-sectional study design of our study which does not enable identifying cause–effect associations. Another limitation is the parental self-reporting of weight, height, gestational weight gain, infant’s birth weight, as well as BF practices and timing of CF introduction by mothers’ some 3–4 years later. The use of the mean educational level as a single indicator for SES is another limitation of the study. Furthermore, the mother’s alcohol consumption during pregnancy which was not investigated as a risk factor for child health may be added to the list of the study limitations. The strengths of our study are the large number of study participants, the inclusion of children from several European countries adding external validity and the standardization of measurement approaches [25,31].

## 5. Conclusions

We conclude that other variables have a greater impact on the risk for childhood obesity than the timing of CF introduction. Therefore, intervention programs for childhood obesity should be conducted, including educating mothers about healthy eating practices and other possible risk factors for overweight.

## Figures and Tables

**Table 1 nutrients-13-01199-t001:** Characteristics of participants by country (* χ^2^ test; ** ANOVA).

	Country, *n* (%)							
	Belgium	Bulgaria	Germany	Greece	Poland	Spain	Total	*p*
**Gender**								
Male	600 (53.2)	438 (50.1)	577 (52.3)	841 (51.1)	707 (53.0)	392 (55.0)	3555 (52.3)	0.37 *
Female	528 (46.8)	436 (49.9)	527 (47.7)	806 (48.9)	627 (47.0)	321 (45.0)	3245 (47.7)	
	1128 (100)	874 (100)	1104 (100)	1647 (100)	1334 (100)	713 (100)	6800 (100)	
**Mean birth weight (±SD)**								
	3.34 (0.51)	3.26 (0.53)	3.32 (0.54)	3.14 (0.53)	3.44 (0.55)	3.32 (0.50)	3.29 (0.54)	<0.001 **
**Ponderal index at birth**								
Low	121 (10.7)	142 (16.2)	28 (2.6)	339 (20.7)	754 (59.6)	95 (13.2)	1633 (24.3)	<0.001 *
Normal	914 (81.0)	690 (78.9)	883 (80.8)	1267 (77.2)	499 (39.5)	554 (77.7)	4806 (71.6)	
High	93 (8.3)	42 (4.9)	28 (2.6)	34 (2.1)	12 (0.9)	66 (9.1)	274 (4.1)	
**BMI at month 6**								
Under-/normal	788 (94.9)	445 (90.6)	917 (91.7)	1334 (92.8)	818 (89.1)	545 (92.1)	4897 (92.0)	<0.001 *
Overweight	35 (4.2)	21 (4.3)	65 (6.5)	85 (5.9)	75 (8.2)	41 (6.9)	322 (6.1)	
Obese	7 (0.9)	25 (5.1)	18 (1.8)	17 (1.3)	25 (2.7)	6 (1.0)	98 (1.9)	
**BMI at month 12**								
Under-/normal	607 (94.4)	400 (82.1)	902 (91.0)	1231 (88.2)	749 (82.6)	515 (88.9)	4404 (88.0)	<0.001 *
Overweight	27 (4.2)	60 (12.4)	62 (6.3)	130 (9.3)	131 (14.4)	55 (9.5)	465 (9.3)	
Obese	9 (1.4)	27 (5.5)	27 (2.7)	35 (2.5)	27 (3.0)	9 (1.6)	134 (2.7)	
**BMI categories of preschoolers, *n* (%)**								
Underweight	8 (0.7)	5 (0.6)	4 (0.4)	11 (0.7)	7 (0.5)	2 (0.3)	37 (0.5)	<0.001 *
Normal weight	1059 (93.9)	764 (87.4)	1024 (92.8)	1356 (82.3)	1214 (91.0)	613 (86.0)	6030 (87.8)	
Overweight	47 (4.2)	76 (8.7)	61 (5.5)	200 (12.1)	83 (6.2)	75 (10.5)	542 (8.0)	
Obese	14 (1.2)	29 (3.3)	14 (1.3)	80 (4.9)	30 (2.2)	23 (3.2)	190 (2.8)	
**SES, *n* (%)**								
Low SES	453 (40.2)	124 (14.2)	243 (22.0)	790 (48.0)	445 (33.4)	290 (40.7)	2345 (34.5)	<0.001 *
Medium SES	341 (30.2)	300 (34.3)	388 (35.1)	448 (27.2)	395 (29.6)	256 (35.9)	2128 (31.3)	
High SES	334 (29.6)	450 (51.5)	473 (42.8)	409 (24.8)	494 (37.0)	167 (23.4)	2327 (34.2)	
	1128 (100)	874 (100)	1104 (100)	1647 (100)	1334 (100)	713 (100)	6800 (100)	
**Mother’s age—mean (±SD)**								
	33.7 (4.7)	33.9 (4.4)	35.7 (5.1)	37.1 (4.4)	34.5 (4.3)	37.7 (4.6)	35.4 (4.7)	<0.001 **
**BMI categories, *n* (%)**								
Under-/normal	755 (70.4)	667 (78.9)	727 (70.9)	1101 (70.1)	1011 (78.6)	503 (74.2)	4764 (73.5)	<0.001 *
Overweight	217 (20.2)	133 (15.7)	213 (20.8)	328 (20.9)	204 (15.9)	134 (19.8)	1229 (19.0)	
Obese	100 (9.3)	45 (5.3)	86 (8.4)	142 (9.0)	72 (5.6)	41 (6.0)	486 (7.5)	
**Tobacco use during pregnancy**								
No smoking	1011 (90.7)	687 (79.8)	956 (89.2)	1340 (82.7)	1220 (93.2)	574 (81.0)	5788 (86.6)	
Smoking, 2nd trimester	2 (0.2)	12 (1.4)	4 (0.4)	40 (2.5)	1 (0.1)	2 (0.3)	61 (0.9)	<0.0001 *
Smoking, 1st and 3rd trimester	6 (0.5)	46 (5.3)	29 (2.7)	54 (3.3)	29 (2.2)	25 (3.5)	189 (2.8)	
Smoking throughout pregnancy	96 (8.6)	116 (13.5)	83 (7.7)	187 (11.5)	59 (4.5)	108 (15.2)	649 (9.7)	

**Table 2 nutrients-13-01199-t002:** Infant feeding practices among pre-school children from the six countries, participating in the ToyBox-study.

Infant-Feeding Practice	Country, *n* (%)							*p*
	Belgium	Bulgaria	Germany	Greece	Poland	Spain	Total	
**Exclusive breastfeeding at** **4–6 months of age, *n* (%)**	32 (2.8)	47 (5.4)	163 (14.8)	44 (2.7)	105 (7.9)	37 (5.2)	428 (6.3)	<0.001
**Ever breastfed rate, *n* (%)**	751 (66.7)	811 (92.8)	928 (84.1)	1418 (86.1)	1263 (94.7)	606 (85.0)	5777 (85.0)	<0.001
**Duration of BF** **(median; months; IQR)**	4 (2–6)	5 (3–9)	7 (4–11)	3 (2–6)	9 (4–13)	5 (2–9)	5 (2–9)	<0.001
**Continued BF rate (>12 months)**	31 (3.9)	87 (10.3)	134 (12.1)	84 (5.9)	347 (26.3)	95 (15.7)	778 (12.8)	<0.001
**Introduction of tea** **(median; months; IQR)**	3 (2–4)	2 (2–4)	3 (1–6)	3 (1–6)	3 (2–5)	3 (2–6)	3 (2–5)	<0.001
**Introduction of fruit juices (Median; months; IQR)**	6 (4–12)	4 (3–6)	8 (6–13)	8 (6–13)	6 (5–7)	6 (5–8)	6 (5–8)	<0.001
**Introduction of SF, months (mean ± SD)**	4.6 ± 1.8	6.6 ± 2.0	6.3 ± 1.8	5.8 ± 1.2	5.8 ± 1.6	5.6 ± 1.5	5.8 ± 1.7	<0.001 *
**Introduction of SF at** **1–3 months of age, *n* (%)**	197 (17.5)	19 (2.2)	18 (1.6)	15 (0.8)	15 (1.1)	16 (2.2)	279 (4.1)	<0.001 ^†^
**Introduction of SF at** **4–6 months of age, *n* (%)**	839 (74.4)	485 (55.5)	694 (63.9)	1385 (84.1)	995 (74.6)	596 (83.6)	4994 (73.4)	
**Introduction of SF at** **7–12 months of age, *n* (%)**	92 (8.1)	370 (42.3)	392 (35.5)	248 (15.1)	324 (24.3)	101 (14.2)	1527 (22.5)	
**Exclusive BF at 4–6 months of age + introduction of SF and BF < 12 months**	32 (2.84)	47 (5.38)	163 (14.76)	44 (2.67)	105 (7.87)	37 (5.19)	428 (6.29)	<0.001
**Exclusive BF at 4–6 months of age + introduction of SF and BF ≥ 12 months**	15 (1.15)	24 (2.90)	74 (7.86)	18 (1.09)	142 (10.64)	18 (2.52)	225 (3.31)	<0.001

* Introduction of solid foods is significantly different with exception of the following comparisons: Greece and Spain (*p* = 0.13); Greece and Poland (*p* = 0.9); Poland and Spain (*p* = 0.18) (ANOVA); ^†^
*p*-value of the ^-^χ^2^ test; EBF—exclusive breastfeeding; SF—solid foods; IQR—interquartile range.

**Table 3 nutrients-13-01199-t003:** Breastfeeding practices and weight status of children (χ^2^ and independent samples *t*-test).

	Weight at Month 6			Weight at Month 12			Weight, Preschoolers		
	Under-/ Normal	Overweight	Obese	Under-/ Normal	Overweight	Obese	Under-/ Normal	Overweight	Obese
EBF at 0–3 months of age; *n* (%)	1672 (91.7)	116 (6.4)	35 (1.9)	1557 (88.6)	163 (9.3)	37 (2.1)	2104 (90.8)	170 (7.3)	43 (1.9)
EBF at 4–6 months of age; *n* (%)	304 (90.2)	23 (6.8)	10 (3.0)	299 (88.7)	29 (8.6)	9 (2.7)	392 (91.6)	29 (6.8)	7 (1.6)
Introduction of CF at 0–3 months of age; *n* (%)	187 (93.5)	7 (3.5)	6 (3.0)	147 (92.5)	7 **(4.4) ***	5 (3.1)	262 (93.9)	12 **(4.3) ***	5 **(1.8) ***
Introduction of CF at 4–6 months of age; *n* (%)	3655 (92.5)	241 (6.1) *	54 (1.4) *	3310 (88.4)	349 **(9.3) ***	86 (2.3)	4447 (89.1)	401 **(8.0) ***	145 **(2.9) ***
Introduction of CF at 7–12 months of age; *n* (%)	1005 (90.0)	74 **(6.6) ***	38 **(3.4) ***	947 (86.2)	109 (9.9)	43 (3.9)	1358 (88.9)	129 (8.4)	40 (2.6)

EBF—exclusive breastfeeding; significant comparisons of the prevalence of overweight and obesity (independent samples *t*-test) at 6 months of age: * introduction of CF (complementary foods) at 4–6 and ≥ 7 months of age—t = 2.71; *p* < 0.01; at 12 months of age: * introduction of CF at 4–6 and 0–3 months of age—t = 4.32; *p* < 0.001; pre-school age: * introduction of CF at 4–6 and 0–3 months of age—t = 2.98; *p* < 0.01.

**Table 4 nutrients-13-01199-t004:** Breastfeeding practices and weight status of preschool children by country (χ^2^).

Country		Weight, Preschoolers, *n* (%)		
		Under-/Normal Weight	Overweight	Obese
Belgium	EBF at 0–3 months of age	304 (93.3)	17 (5.2)	5 (1.5)
	EBF at 4–6 months of age	29 (90.6)	1 (3.1)	2 (6.3)
	Introduction of CF at 0–3 months of age	187 (94.9)	8 (4.1)	2 (1.0) *
	Introduction of CF at 4–6 months of age	801 (95.5)	32 (3.8)	6 (0.7) **
	Introduction of CF at 7–12 months of age	79 (85.9)	7 (7.6)	6 (6.5) *
Bulgaria	EBF at 0–3 months of age	190 (91.8)	16 (7.7)	1 (0.5)
	EBF at 4–6 months of age	43 (91.5)	4 (8.5)	0
	Introduction of CF at 0–3 months of age	19 (100)	0	0
	Introduction of CF at 4–6 months of age	416 (85.8)	49 (10.1)	20 (4.1)
	Introduction of CF at 7–12 months of age	334 (90.3)	27 (7.3)	9 (2.4)
Germany	EBF at 0–3 months of age	458 (94.0)	23 (4.7)	6 (1.2)
	EBF at 4–6 months of age	158 (96.9)	4 (2.5)	1 (0.6)
	Introduction of CF at 0–3 months of age	16 (88.9)	1 (5.6)	1 (5.6)
	Introduction of CF at 4–6 months of age	643 (92.8)	43 (6.2)	7 (1.0)
	Introduction of CF at 7–12 months of age	369 (94.1)	17 (4.3)	6 (1.5)
Greece	EBF at 0–3 months of age	305 (84.7)	44 (12.2)	11 (3.1)
	EBF at 4–6 months of age	36 (81.8)	6 (13.6)	2 (4.5)
	Introduction of CF at 0–3 months of age	12 (85.7)	0	2 (14.3)
	Introduction of CF at 4–6 months of age	1156 (83.5)	164 (11.8)	65 (4.7)
	Introduction of CF at 7–12 months of age	199 (80.2)	36 (14.5)	13 (5.2)
Poland	EBF at 0–3 months of age	575 (92.3)	37 (5.9)	11 (1.8)
	EBF at 4–6 months of age	95 (90.5)	9 (8.6)	1 (1.0)
	Introduction of CF at 0–3 months of age	14 (93.3)	1 (6.7)	0
	Introduction of CF at 4–6 months of age	916 (92.1)	53 (5.3) *	26 (2.6)
	Introduction of CF at 7–12 months of age	291 (89.8)	29 (9.0) *	4 (1.2)
Spain	EBF at 0–3 months of age	272 (86.6)	33 (10.5)	9 (2.9)
	EBF at 4–6 months of age	31 (83.8)	5 (13.5)	1 (2.7)
	Introduction of CF at 0–3 months of age	14 (87.5)	2 (12.5)	0
	Introduction of CF at 4–6 months of age	515 (86.4)	60 (10.1)	21 (3.5)
	Introduction of CF at 7–12 months of age	86 (85.1)	13 (12.9)	2 (2.0)

Significant comparisons of the prevalence of overweight and obesity in preschool age (independent samples *t*-test): Belguim: obesity; * introduction of CF (complementary foods) at 0–3 and 7–12 months of age (t = 2.06; *p* < 0.02); ** introduction of CF at 4–6 and 7–12 months of age (t = 2.27; *p* < 0.001); Poland: overweight; * introduction of CF at 4–6 and ≥ 7 months of age (t = 2.12; *p* = 0.03).

**Table 5 nutrients-13-01199-t005:** Maternal characteristics associated with non-compliance to the recommendation for introduction of complementary foods (CF) at 4–6 months of age.

	Introduction of SF at 4–6 Months of Age (*n* = 4266)	
	OR (95% CI), *p*	
Smoking habits throughout pregnancy ^1^		
No smoking	1 (reference)	
Smoking, 2nd trimester	1.50 (0.75–3.01)	0.25
Smoking, 1st and 3rd trimesters	1.07 (0.75–1.53)	0.71
Smoking throughout pregnancy	1.22 (0.98–1.52)	0.08
BMI before pregnancy ^2^		
Underweight	1.04 (0.85–1.29)	0.68
Normal weight	1 (reference)	
Overweight	1.16 (0.97–1.39)	0.11
Obese	1.20 (0.89–1.62)	0.24
SES ^3^		
Low	1.25 (1.08–1.45)	0.002
Medium	1.11 (0.96–1.28)	0.17
High	1 (reference)	

^1^ Adjusted for age and BMI before pregnancy, country and SES. ^2^ Adjusted for age before pregnancy, smoking habits during pregnancy, country and SES. ^3^ Adjusted for age and BMI before pregnancy, smoking habits during pregnancy, country.

## Data Availability

The data presented in this study are available on request from the corresponding author. The data are not publicly available due to restrictions of informed consent and the requirement of IRB review and approval.

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
