# Peer review of "Complementary Feeding and Overweight in European Preschoolers: The ToyBox-Study"

_nutrients, 2021, doi:10.3390/nu13041199_

Round 1

Reviewer 1 Report

Please add the date in these following sentences (2018? 2019?)

[Moreover, health  expenditures for the adult population are constantly increasing currently €70 billion per year in Europe and $342.2 billion in the US. Direct medical costs related to childhood obesity alone are approximately $14 billion and they are expected to rise significantly, especially because today’s obese children are likely to become tomorrow’s obese adults]

The alcohol consuming during the pregnancy is a clear risk factor for child and mother health; so if it is possible, I suggest to introduce in the analysis this determinant. If is not possible to add this information, I suggest to explain it in the discussion and in the section “limits of the study”.

Author Response

Please add the date in these following sentences (2018? 2019?)

[Moreover, health  expenditures for the adult population are constantly increasing currently €70 billion per year in Europe and $342.2 billion in the US. Direct medical costs related to childhood obesity alone are approximately $14 billion and they are expected to rise significantly, especially because today’s obese children are likely to become tomorrow’s obese adults]

Response: 

We thank the Reviewer for the requirement and the dates are added as follow:

Moreover, health expenditures for the adult population are constantly increasing currently €70 billion per year in Europe (2017) and $342.2 billion in the US (2013). Direct medical costs related to childhood obesity alone are approximately $14 billion in 2013 and they are expected to rise significantly, especially because today’s obese children are likely to become tomorrow’s obese adults.

The alcohol consuming during the pregnancy is a clear risk factor for child and mother health; so if it is possible, I suggest to introduce in the analysis this determinant. If is not possible to add this information, I suggest to explain it in the discussion and in the section “limits of the study”.

Response: We thank the Reviewer for the suggestion and we agree that the alcohol consumption during the pregnancy is a risk factor for child and mother’s health but it had not been included in the list of investigated variables in the ToyBox Study.

Reviewer 2 Report

I appreciate the opportunity to read and review this article. A very interesting work to deepen our knowledge of the variables that can prevent childhood overweight and obesity. I think the work is well prepared and the article well written. The weakest point of the study is that it is not a longitudinal study, but based on the information that is remembered by the parents, information that, in addition to including errors in the remembered data, may be influenced by the child's current weight status. This matter mentioned briefly in the limitations of the work, should be analyzed in more detail in the main text. It should also be explained in the methodology why variables on the family's lifestyle and diet have not been collected at the time of considering weight at pre-school age, given that it could be a determining variable of weight status. This is mentioned very briefly in the limitations, but the methodology should justify why the study of this variable is waived.

Author Response

I appreciate the opportunity to read and review this article. A very interesting work to deepen our knowledge of the variables that can prevent childhood overweight and obesity. I think the work is well prepared and the article well written. The weakest point of the study is that it is not a longitudinal study, but based on the information that is remembered by the parents, information that, in addition to including errors in the remembered data, may be influenced by the child's current weight status. This matter mentioned briefly in the limitations of the work, should be analyzed in more detail in the main text. It should also be explained in the methodology why variables on the family's lifestyle and diet have not been collected at the time of considering weight at pre-school age, given that it could be a determining variable of weight status. This is mentioned very briefly in the limitations, but the methodology should justify why the study of this variable is waived.

Response:

We thank the Reviewer for the suggestion. The information about the families' lifestyle and dietary habits had been collected using the validated questionnaires (FFQ, physical activity and sedentary behavior). Parents/caregivers were asked to describe the children’s usual food and beverage habits over the last 1–2 months in an FFQ for young children, based on a previously validated FFQ developed by Huybrechts et al. Results regarding this topic are presented in other papers:  

  1. Pinket AS, De Craemer M, Maes L, De Bourdeaudhuij I, Cardon G, Androutsos O, Koletzko B, Moreno L, Socha P, Iotova V, Manios Y, Van Lippevelde W. Water intake and beverage consumption of pre-schoolers from six European countries and associations with socio-economic status: the ToyBox-study. Public Health Nutr. 2016 Sep;19(13):2315-25. doi: 10.1017/S1368980015003559. Epub 2015 Dec 18. PMID: 26680732.
  2. Cardon G, De Bourdeaudhuij I, Iotova V, Latomme J, Socha P, Koletzko B, Moreno L, Manios Y, Androutsos O, De Craemer M; ToyBox-study group. Health Related Behaviours in Normal Weight and Overweight Preschoolers of a Large Pan-European Sample: The ToyBox-Study. PLoS One. 2016 Mar 7;11(3):e0150580. doi: 10.1371/journal.pone.0150580. PMID: 26950063; PMCID: PMC4780703.

We will include the following sentence in the methodology section in order to reflect your suggestion: “Data for health related behaviours (dietary habits, physical activity and sedentary behaviour) of preschoolers and their parents had been collected using the validated questionnaires and the results about this topic were presented in other papers [26, 27].

Reviewer 3 Report

Usheva and colleagues have conducted a well designed study on the association between complementary feeding, breastfeeding and overweight/obesity in preschool children and concluded that there were no identified associations between timing of introducing solid food and obesity in childhood. The paper is well written and the aims are clear, with clear presentation of the results and discussion.

Comments

In the abstract conclusion (last sentence) the authors write that 'other factors (e.g. genetic, behaviour) have more impact...' While this may be true, this is not what the findings of this study showed and may not be the best way to end conclude. It may be better to rephrase this to say that other factors may have an impact and perhaps not specify genetic or behavioural.

The introduction and methods are well written and clear. Have the authors used maternal education status as the sole indicator of SES? While it is often a good indicator, it may be worth mentioning this in the limitations section or even re-labelling SES as maternal education.

The results in Table 1 suggest differences across the countries for all variables except gender. Have the authors factored in country in the analyses? Would that impact any of the other results perhaps mentioned in Table 3? I see that smoking was mentioned in this regard in the text as being different in Belgium.

Lines 194-196 is a repeat of lines 184-187.

In the discussion, the authors need to acknowledge the differences across the different countries when making any conclusions, and need to mention this in the limitations. Also, these results are relevant to European countries, but would the authors like to discuss what the implications of this study may be for other populations like those where malnutrition or infectious disease may be more of a problem, like in South Asia or regions in Africa?

In the conclusion, as for the abstract, it may not be ideal to suggest SES, lifestyle and genetic predictors have a greater impact... While this may feature in the discussion, it may be better to either be less definitive, or just mention the results that the study has shown rather than speculate.

Author Response

In the abstract conclusion (last sentence) the authors write that 'other factors (e.g. genetic, behaviour) have more impact...' While this may be true, this is not what the findings of this study showed and may not be the best way to end conclude. It may be better to rephrase this to say that other factors may have an impact and perhaps not specify genetic or behavioural.

Response:

We are very grateful for the Reviewer’s in-depth comments and positive evaluation.

We agree with the suggestion and the conclusion will be changed: “It is likely that other factors than timing of SF-introduction, may have impact on childhood obesity”.

Have the authors used maternal education status as the sole indicator of SES? While it is often a good indicator, it may be worth mentioning this in the limitations section or even re-labelling SES as maternal education.

Response:

We thank the Reviewer for this very accurate remark. 

In this analysis we used “SES” that was based on the mean educational level at municipality/region/neighbourhood level and no other indicator of SES. We agree with the Reviewer that SES cannot be clearly depicted if only education level is used and therefore we will add this as a limitation.:

The using of mean educational level as a single indicator for SES is next limitation of the study.

The results in Table 1 suggest differences across the countries for all variables except gender. Have the authors factored in country in the analyses? Would that impact any of the other results perhaps mentioned in Table 3? I see that smoking was mentioned in this regard in the text as being different in Belgium.

Response: 

A new table (Table 4) is added in the manuscript text with overweight prevalence data per country.

Lines 194-196 is a repeat of lines 184-187.

Response: 

We are thankful to the Reviewer for noticing this technical mistake that is now corrected.

In the discussion, the authors need to acknowledge the differences across the different countries when making any conclusions, and need to mention this in the limitations. Also, these results are relevant to European countries, but would the authors like to discuss what the implications of this study may be for other populations like those where malnutrition or infectious disease may be more of a problem, like in South Asia or regions in Africa?

Response:

Our results which hopefully will be useful for improving of the effectiveness of childhood obesity prevention programs in Europe, can also be utilised in low developed and developing countries. Although malnutrition is still a major challenge across the African continent, the largest growth of obesity among 5- to 19-year-olds in the world between 1975 and 2016 was observed in southern Africa (about 400% per decade)[45].

In the conclusion, as for the abstract, it may not be ideal to suggest SES, lifestyle and genetic predictors have a greater impact... While this may feature in the discussion, it may be better to either be less definitive, or just mention the results that the study has shown rather than speculate.

Response:

We agree with the suggestion and the conclusion will be changed: “It is likely that other factors than timing of SF-introduction, may have impact on childhood obesity”.

Reviewer 4 Report

The study focuses on the association of complementary feeding, breastfeeding and overweight in preschool children. The study is interesting.

However, some methodological elements are missing, in particular statistical analysis.
Have you estimated the number of subjects to be included in the analysis?
Provide a schematic or diagram to explain the study and patient follow-up?

Verify the statistical parameters' distribution and apply the appropriate tests in univariate analysis (comparison or correlation).
In the logistic regression model, how did you select the variables to be included in the model? Justify? Did you test for nonlinear links? Interactions?
Did you validate and calibrate the model?
Present the ORs with the CIs estimated by bootstrapping!

Author Response

The study focuses on the association of complementary feeding, breastfeeding and overweight in preschool children. The study is interesting. However, some methodological elements are missing, in particular statistical analysis.

Have you estimated the number of subjects to be included in the analysis?
Provide a schematic or diagram to explain the study and patient follow-up?

Response:

We thank the Reviewer for the questions and suggestions.

The methodological issues of the ToyBox study design are represented in other published articles and more specifically:  Manios Y, Androutsos O., Katsarou C et al. , Designing and implementing a kindergarten-based, family-involved intervention to prevent obesity in early childhood. The Toybox-study. Obes Rev, 2014. 15 (Suppl. 3): p. 5-13.                             The above was already mentioned in the muniscript text: Detailed information about the ToyBox-study design has been previously reported [24].” (Lines 89-90)

Verify the statistical parameters' distribution and apply the appropriate tests in univariate analysis (comparison or correlation). In the logistic regression model, how did you select the variables to be included in the model? Justify? Did you test for nonlinear links? Interactions?
Did you validate and calibrate the model?
Present the ORs with the CIs estimated by bootstrapping!

Response:

Distribution of the analyzed variables were tested for normality. Regression model includes the several confounding risk factors based on the published literature (mother’s age and BMI before pregnancy, smoking habits during pregnancy, SES). Child’s age, gender, and country were treated as covariates.

The model was validate on the basis of Chi-square coefficient and significance from Omnibus test of Model (p<0.05); Cox&Snell R2 and Nagelkerke R2 from the Model Summary, as well as the overall predicted correct percentage  from the Classification table in SPSS. We did not calibrate the model.

Bootstrapping was not considered for the statistical processing of the sampled data. 

Round 2

Reviewer 4 Report

I think the authors' response deserves to be a little more detailed. Could you please answer, with your justifications, more clearly and precisely to the various questions?

Verify the statistical parameters' distribution and apply the appropriate tests in univariate analysis (comparison or correlation). In the logistic regression model, how did you select the variables to be included in the model? Justify? Did you test for nonlinear links? Interactions?
Did you validate and calibrate the model? Justify ?

Author Response

I think the authors' response deserves to be a little more detailed. Could you please answer, with your justifications, more clearly and precisely to the various questions?

Verify the statistical parameters' distribution and apply the appropriate tests in univariate analysis (comparison or correlation). In the logistic regression model, how did you select the variables to be included in the model? Justify? Did you test for nonlinear links? Interactions?
Did you validate and calibrate the model? Justify ?

Response:

We thank the Reviewer for the in-depth scrutinizing of the provided statistical analysis. We hope very much that our current explanations would be deemed acceptable. In particular, we propose the text below with these  clarifications to be included in the manuscript.

Statistical analyses

Normal distribution of variables was tested with Shapiro-Wilk tests. Continuous variables are presented as mean ± standard deviation in case they were normally distributed (e.g. age of preschoolers, age of mothers, introduction of solid foods) and by median and IQR (Interquartile range) for non-normally distributed variables (duration of breastfeeding, introduction of tea, introduction of fruit juice). Statistical analysis of parameters' distribution of the original samples by country was not applied as their number is small (n=6). Post-sampling or bootstrapping have not been considered.

Categorical variables were analysed by χ2-test regarding country and children’s BMI categories. Independent Sample t-test was applied for comparison of means and percentages from 2 samples while the One-way ANOVA (Analysis of variance) for means of more than 2 samples (birth weight; mother’s age).

Comparison of medians was performed by the Median test. The association of feeding practices and children’s BMI as well as with mother’s characteristics was determined by the correlation analysis. Logistic regression analysis with 95% confidence intervals (CIs) was used to estimate the odds of being overweight/obese (dependent variable) in relation to different infant feeding practices. Results were adjusted for mother’s age and BMI before pregnancy, SES, smoking habits during pregnancy and country. In order to quantify the probability of complying with current recommendation for introduction of CF at 4-6 months, logistic regression analysis was performed and adjusted for mother’s age and BMI before pregnancy, SES, smoking habits during pregnancy and country. Compliance to recommendations for introduction of CF at 4-6 months (yes/no) was considered as dependent variable.

In the logistic regression models, the variables were selected based on their relevance for the research topic and being tested for absence of collinearity, hence the presented model coefficients correspond to variables with no significant impact as well. Thus we can reach a conclusion about existence or not of meaningful links. The regression analyses of the current data were targeted at identification of the relevant links among the study variables, but not at constituting the universal model, which maybe applied for analysis of other populations or for establishing of new theories. Data was analysed using the Statistical Package for Social Sciences (IBM SPSS v. 20, Chicago, IL, USA). Level of significance was set at p < 0.05.